# Lipid nanodisc scaffold and size alter the structure of a pentameric ligand-gated ion channel

Vikram Dalal[1,5], Mark J. Arcario [1,5], John T. Petroff II[1], Brandon K. Tan[1], Noah M. Dietzen[1], Michael J. Rau [2], James A. J. Fitzpatrick[2], Grace Brannigan[3,4] & Wayland W. L. Cheng [1] ✉

Lipid nanodiscs have become a standard tool for studying membrane proteins, including using single particle cryo-electron microscopy (cryo-EM). We find that reconstituting the pentameric ligand-gated ion channel (pLGIC), *Erwinia* ligand-gated ion channel (ELIC), in different nanodiscs produces distinct structures by cryo-EM. The effect of the nanodisc on ELIC structure extends to the extracellular domain and agonist binding site. Additionally, molecular dynamic simulations indicate that nanodiscs of different size impact ELIC structure and that the nanodisc scaffold directly interacts with ELIC. These findings suggest that the nanodisc plays a crucial role in determining the structure of pLGICs, and that reconstitution of ion channels in larger nanodiscs may better approximate a lipid membrane environment.

Lipid nanodiscs are routinely used for the reconstitution of membrane proteins for structure determination by single particle cryo-EM. Multiple types of nanodiscs are available including the membrane scaffold protein (MSP) nanodiscs[1] which can be modified to produce a circularized scaffold[2,3], saposin nanodiscs[4,5], and polymer nanodiscs which use synthetic polymers such as styrene maleic acid (SMA)[6]. We will refer to these collectively as nanodiscs, distinguishing the original Nanodisc technology as MSP nanodiscs[7]. While it is often assumed that nanodiscs mimic the environment of a model lipid bilayer, studies of the bilayer properties of empty MSP nanodiscs (i.e., nanodiscs without a membrane protein) indicate that nanodisc size has complex effects on membrane thickness, order and stiffness[8–11]. Furthermore, a recent cryo-EM structure of SARS-CoV-2 ORF3a in nanodiscs showed direct interactions of the scaffold with the membrane protein[12]. Therefore, it is possible that the nanodisc scaffold can influence membrane protein structure by altering the properties of the lipid bilayer or directly interacting with the protein of interest. Understanding this is essential to define functionally-relevant conformations and to characterize membrane effects on protein structure.

Structures of pentameric ligand-gated ion channels (pLGICs) in nanodiscs have revolutionized the structural biology and pharmacology of these ion channels[13]. However, pLGICs are sensitive to their membrane environment[14,15], which suggests that interactions between the nanodisc and protein might affect protein conformation. Structures of several pLGICs have been determined in different nanodiscs including the glycine receptor (GlyR) in MSP1E3D1 and SMA nanodiscs[16–18], and the GABA(A) receptor (GABA$_A$R)[19,20] and muscle-type nicotinic acetylcholine receptor (nAchR)[21,22] in MSP2N2 and saposin nanodiscs. In the cases of GlyR and GABA$_A$R, distinct conformations were observed in the different nanodiscs. However, in both cases, different lipids or protein constructs were used, such that the impact of the nanodisc per se on protein structure was unclear.

Here, we report differences in the structure of the pLGIC, ELIC (*Erwinia* ligand-gated ion channel), in different nanodiscs and perform MD simulations to explore the underlying mechanism. ELIC is one of a few pLGICs for which there is an activated, open-channel structure, and is therefore a useful structural model[23]. The results indicate that the nanodisc alters the structure of ELIC, and suggest that larger

[1]Department of Anesthesiology, Washington University School of Medicine, Saint Louis, MO, USA. [2]Center for Cellular Imaging, Washington University School of Medicine, Saint Louis, MO, USA. [3]Center for Computational and Integrative Biology, Rutgers University, Camden, NJ, USA. [4]Department of Physics, Rutgers University, Camden, NJ, USA. [5]These authors contributed equally: Vikram Dalal, Mark J. Arcario. ✉e-mail: wayland.cheng@wustl.edu

nanodiscs may better approximate a native-like membrane environment for structural studies.

## Results

### Structures of ELIC in different nanodiscs

To test the impact of nanodiscs on ELIC structure, we reconstituted ELIC in four different nanodiscs: SMA[6], saposin[4], and the circularized MSP nanodiscs, spMSP1D1 and spNW15[3]. The "sp" in spMSP1D1 refers to the Spycatcher-Spytag technology used to generate the circularized MSP scaffold: spMSP1D1 produces a nanodisc with a diameter of 11 nm and spNW15 15 nm[3]. A 2:1:1 ratio of POPC (1-palmitoyl-2-oleoyl-phosphatidylcholine): POPE (1-palmitoyl-2-oleoyl-phosphatidylethanolamine): POPG (1-palmitoyl-2-oleoyl-phosphatidylglycerol) (hereafter called 2:1:1 lipids) were used in all nanodisc preparations. Single particle cryo-EM of ELIC in these nanodiscs in the presence of the agonist, propylamine (50 mM), yielded structures with an overall resolution of 3.71 Å for SMA, 3.28 Å for saposin, 3.12 Å for spMSP1D1, and 3.17 Å for spNW15 (Supplementary Table 1, Supplementary Figs. 1 and 2). We previously reported structures of ELIC in MSP1E3D1 nanodiscs with 2:1:1 lipid in the absence and presence of agonist which are resting and agonist-bound non-conducting conformations, respectively, as well as a structure of a non-desensitizing mutant, ELIC5 (P254G/C300S/V261Y/G319F/I320F), which is a putative open-channel conformation[23]. These structures will be referred to as apo-MSP1E3D1$_{ELIC}$ (WT ELIC without agonist in MSP1E3D1), MSP1E3D1$_{ELIC}$ (WT ELIC with agonist in MSP1E3D1) and MSP1E3D1$_{ELIC5}$ (ELIC5 with agonist in MSP1E3D1). In this study, we will compare these structures with the agonist-bound structures of ELIC in SMA, saposin, spMSP1D1, and spNW15 (hereafter called SMA$_{ELIC}$, saposin$_{ELIC}$, spMSP1D1$_{ELIC}$ and spNW15$_{ELIC}$) (Fig. 1). To assess the impact of nanodiscs on the unliganded structure, we also obtained a structure of ELIC without agonist in spMSP1D1 (apo-spMSP1D1$_{ELIC}$) at an overall resolution of 3.17 Å. To assess the impact of nanodiscs on the ELIC5 open-channel structure, we obtained a structure of ELIC5 with agonist in spNW15 (spNW15$_{ELIC5}$) at an overall resolution of 3.4 Å (Fig. 1, Supplementary Table 1 and Supplementary Figs. 1 and 2). No significant differences were observed between the apo structures and ELIC5 structures in different nanodiscs, so we focused our analysis on the WT agonist-bound structures.

One factor that may influence the structure of ELIC is nanodisc size. The diameter of the cryo-EM density for the nanodisc in each structure was estimated from low-pass filtered maps as previously described[19], producing diameters of 9.2 nm for SMA$_{ELIC}$, 9.3 nm for MSP1E3D1$_{ELIC}$, 9.4 nm for saposin$_{ELIC}$, 9.9 nm for spMSP1D1$_{ELIC}$, and 8.4 nm for spNW15$_{ELIC}$ (Fig. 1). The nanodisc density for spNW15$_{ELIC}$ was smaller than expected possibly because the position of ELIC within the nanodisc is more heterogeneous. We, therefore, also assessed nanodisc size using size exclusion chromatography (Supplementary Fig. 3). The effective diameter of ELIC, which depends partially on nanodisc size, agrees with the trend observed in the structures except spNW15$_{ELIC}$. spNW15$_{ELIC}$ produced the largest nanodisc particle as expected for this circularized scaffold which forms 15 nm empty nanodiscs. Therefore, among agonist-bound ELIC structures, nanodisc size varied in the order: SMA > MSP1E3D1 > saposin > spMSP1D1 > spNW15.

### Effect of nanodiscs on the transmembrane domain of ELIC

The structures of agonist-bound ELIC differ in the various nanodiscs. We first examined the pore, which is often used to assign putative functional states to pLGIC structures. The pore-lining residues of the transmembrane helix M2, 16′ (F247) and 9′ (L240), in apo-MSP1E3D1$_{ELIC}$ and apo-spMSP1D1$_{ELIC}$ form hydrophobic constrictions consistent with a resting non-conducting conformation, while the MSP1E3D1$_{ELIC5}$ and spNW15$_{ELIC5}$ pores are open (Fig. 2a, Supplementary Fig. 4)[23,24]. Activation and opening of the channel involves a tilting of the top of M2 away from the pore axis and partial loss of the helix, such that 16′ and 9′ turn away from the pore (Figs. 2a, b). Comparison of the agonist-bound structures of ELIC in the different nanodiscs also shows progressive tilting of the top of M2 away from the pore axis in the order: SMA$_{ELIC}$ < saposin$_{ELIC}$~MSP1E3D1$_{ELIC}$~spMSP1D1$_{ELIC}$ < spNW15$_{ELIC}$. This is evident in the mean distance between M2 Cα

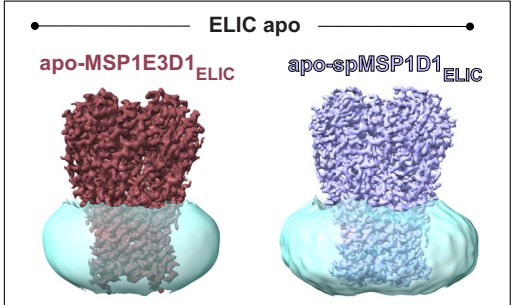

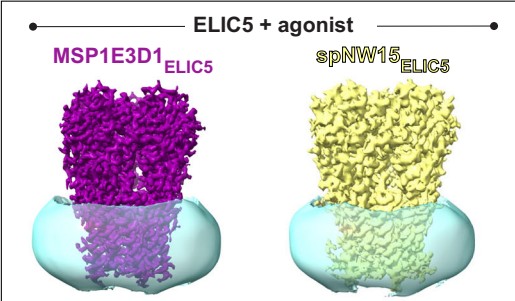

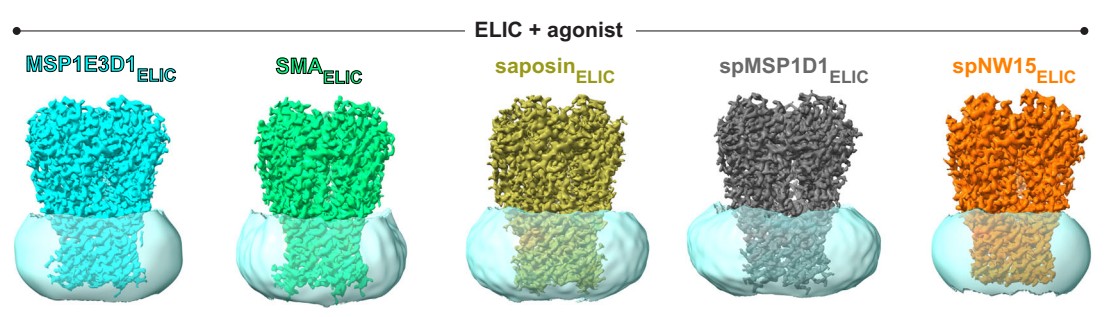

**Fig. 1 | Cryo-EM structures of ELIC in different nanodiscs.** Each structure is named according to whether agonist is absent (apo) or present (no label), followed by the type of scaffold used, followed by the ion channel protein (ELIC or ELIC5) in subscript. The colors used for each structure from PyMOL 2.5.2 are: raspberry for apo-MSP1E3D1$_{ELIC}$, light blue for apo-spMSP1D1$_{ELIC}$, deep purple for MSP1E3D1$_{ELIC5}$, pale yellow for spNW15$_{ELIC5}$, cyan for MSP1E3D1$_{ELIC}$, lime green for SMA$_{ELIC}$, deep olive for saposin$_{ELIC}$, gray for spMSP1D1$_{ELIC}$, and orange for spNW15$_{ELIC}$. Apo-MSP1E3D1$_{ELIC}$ is EMD-27217, MSP1E3D1$_{ELIC}$ is EMD-27218, and MSP1E3D1$_{ELIC5}$ is EMD-27220. ELIC and ELIC5 densities are shown using the sharpened cryo-EM maps, and the nanodisc density is shown using the unsharpened maps low-pass filtered at 8 Å.

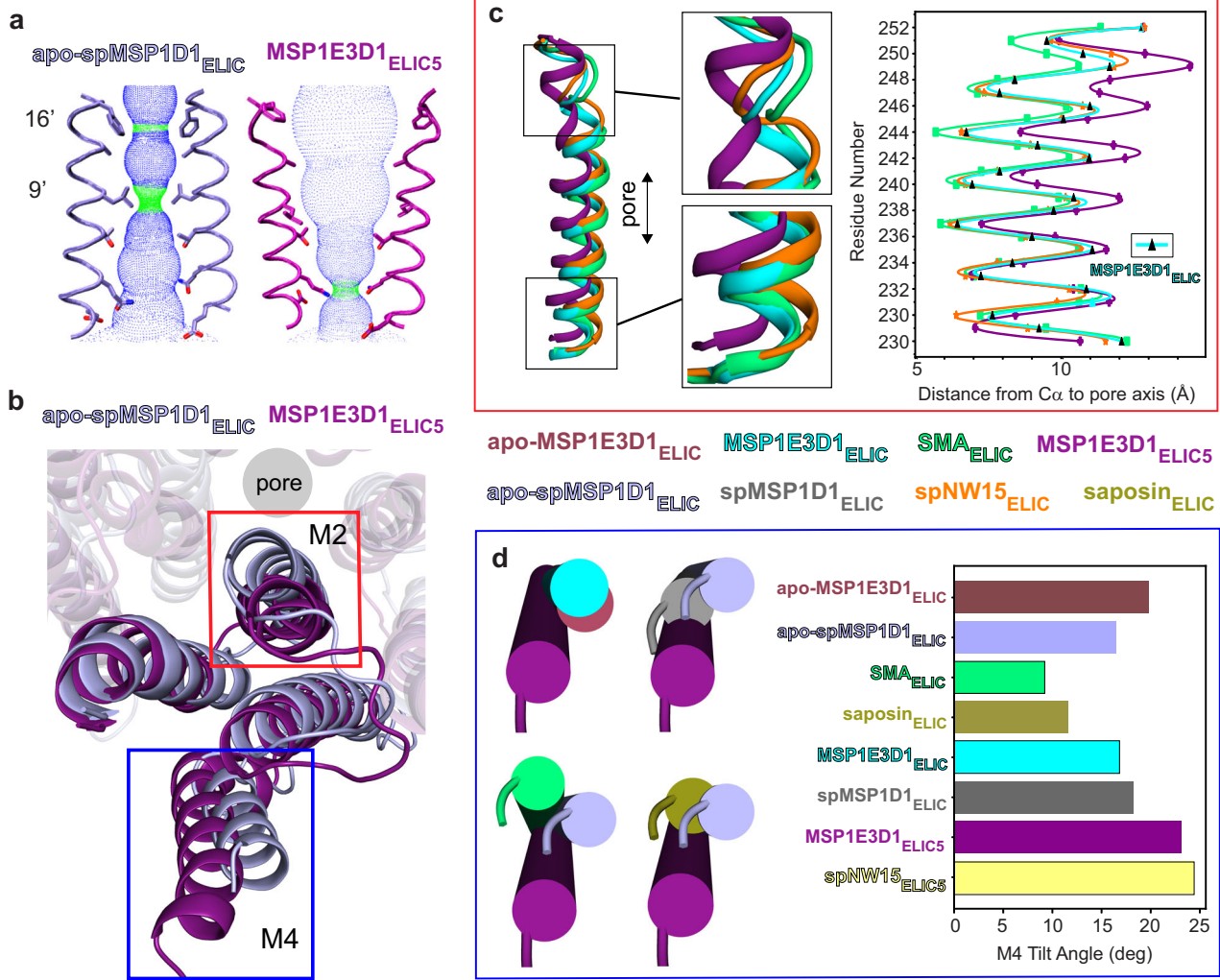

**Fig. 2 | Transmembrane domain (TMD) of ELIC in different nanodiscs. a** Pore profile of apo-spMSP1D1$_{ELIC}$ (light blue) and MSP1E3D1$_{ELIC5}$ (deep purple) showing constriction at 16' and 9' in the apo structure. **b** Top view of a TMD subunit of apo-spMSP1D1$_{ELIC}$ and MSP1E3D1$_{ELIC5}$ with red and blue boxes indicating changes in M2 (**c**) and M4 and (**d**), respectively. **c** Side view of M2 with zoomed-in views of the top and bottom of the helix, and a plot of the M2 Cα-atom distance to the pore axis for SMA$_{ELIC}$ (lime green), MSP1E3D1$_{ELIC}$ (cyan), spNW15$_{ELIC}$ (orange), and MSP1E3D1$_{ELIC5}$ (deep purple). For clarity, MSP1E3D1$_{ELIC}$ is represented as cyan with black triangles in the M2 Cα-atom distance plot. **d** Top views of M4, and graph of the M4 tilt angle relative to the pore axis for apo-MSP1E3D1$_{ELIC}$ (raspberry), apo-spMSP1D1$_{ELIC}$ (light blue), SMA$_{ELIC}$ (lime green), saposin$_{ELIC}$ (deep olive), MSP1E3D1$_{ELIC}$ (cyan), spMSP1D1$_{ELIC}$ (gray), MSP1E3D1$_{ELIC5}$ (deep purple), and spNW15$_{ELIC5}$ (pale yellow). All images are from a superposition of the TMD.

atoms and the pore axis, especially residues 249–251 (Fig. 2c, Supplementary Figs. 4 and 5). spNW15$_{ELIC}$ also shows a significant tilting of the bottom of M2 towards the pore axis and the adjacent subunit, narrowing the pore at 2' (Q233) and -2' (E230) (Fig. 2c). Such narrowing at the bottom of the pore is observed in desensitized structures of the GlyR[16,18], and NMR measurements suggest this conformational change is also associated with desensitization in ELIC[25]. However, in spNW15$_{ELIC}$ and all other agonist-bound WT structures, the position of 9' is unchanged forming a tight constriction of the ion permeation pathway (Supplementary Fig. 4b). This is not unlike the desensitized structure of α7 nAchR where 9' appears to form the desensitization gate[26,27]; therefore, we suggest that spNW15$_{ELIC}$ is a plausible desensitized structure of ELIC.

Activation of ELIC is also characterized by translation and tilting of the membrane-facing M4 helix away from the pore axis (Fig. 2b). While the cryo-EM density of M4 is generally weaker in agonist-bound ELIC structures[23,28], the backbone structure is clearly appreciated from the unsharpened maps with the exception of spNW15$_{ELIC}$ (Supplementary Fig. 6). Comparison of the agonist-bound structures of WT ELIC in the different nanodiscs shows different orientations of M4. The degree of

M4 tilting relative to the pore axis follows the order: SMA$_{ELIC}$ < saposin$_{ELIC}$ < MSP1E3D1$_{ELIC}$ < spMSP1D1$_{ELIC}$ (Fig. 2d). There is no density for M4 in spNW15$_{ELIC}$ indicating that this helix is structurally heterogeneous in the spNW15 nanodisc (Supplementary Fig. 6). Comparison of M4 between corresponding unliganded and agonist-bound structures in MSP1E3D1 and spMSP1D1 also shows a distinct pattern. The tilt angle of M4 in MSP1E3D1$_{ELIC}$ decreases relative to apo-MSP1E3D1$_{ELIC}$, while the M4 tilt angle of spMSP1D1$_{ELIC}$ increases compared to apo-spMSP1D1$_{ELIC}$ (Fig. 2d).

In summary, the agonist-bound structures of ELIC in various nanodiscs produce structural differences in the TMD especially in M2 and M4. SMA$_{ELIC}$ and spNW15$_{ELIC}$ show the least and greatest agonist-dependent changes, respectively. Consistent with SMA limiting ELIC activation, the top of M1 and M3, and the M2-M3 linker of SMA$_{ELIC}$ are more similar to unliganded structures compared to the other agonist-bound structures (Supplementary Fig. 7).

**Effect of nanodiscs on the extracellular domain of ELIC**
In the extracellular domain (ECD), ELIC activation is characterized by a counter-clockwise rotation of the ECD relative to the TMD. There is

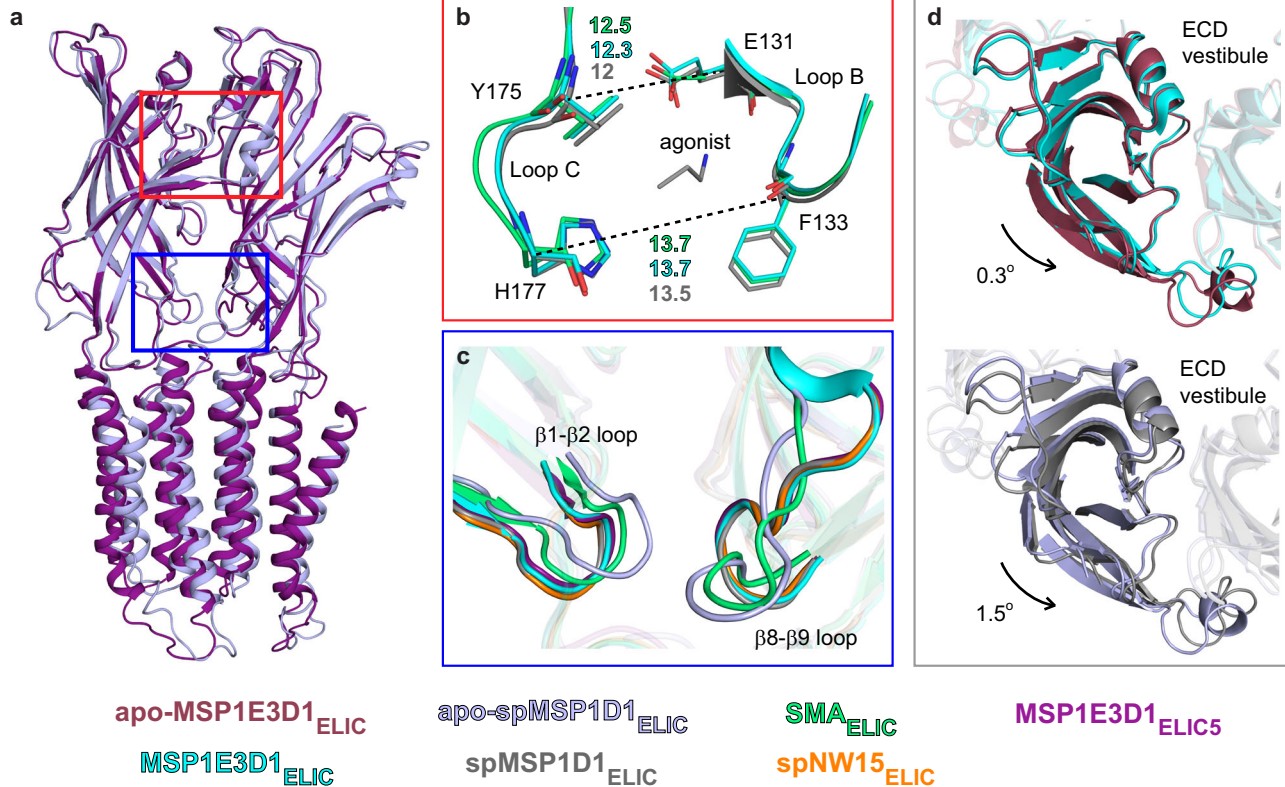

apo-MSP1E3D1_ELIC
MSP1E3D1_ELIC

apo-spMSP1D1_ELIC
spMSP1D1_ELIC

SMA_ELIC
spNW15_ELIC

MSP1E3D1_ELIC5

**Fig. 3 | ECD of ELIC in different nanodiscs scaffolds. a** Global superposition of apo-spMSP1D1_ELIC (light blue) and MSP1E3D1_ELIC5 (deep purple) with red and blue boxes highlighting the agonist binding site (**b**) and the β1-β2 and β8-β9 loops (**c**), respectively. **b** Loops B and C of the agonist binding site with the distance (Å) between the indicated residues shown for SMA_ELIC (lime green), MSP1E3D1_ELIC (cyan), and spMSP1D1_ELIC (gray). Image shows a superposition of the ECD. **c** Global superposition of structures showing side view of β1-β2 and β8-β9 loops for apo-spMSP1D1_ELIC (light blue), SMA_ELIC (lime green), MSP1E3D1_ELIC (cyan), spMSP1D1_ELIC (gray), spNW15_ELIC (orange), and MSP1E3D1_ELIC5 (deep purple). **d** Top view of an ECD subunit showing the counter-clockwise rotation of apo-MSP1E3D1_ELIC (raspberry) compared to MSP1E3D1_ELIC (cyan), and apo-spMSP1D1_ELIC (light blue) compared to spMSP1D1_ELIC (gray). Images show a superposition of the TMD to illustrate rotation of the ECD relative to the TMD.

also contraction of the agonist binding site, and translation of the β1-β2 and β8-β9 loops away from adjacent subunits (Fig. 3a). These agonist-dependent changes are conserved in other pLGICs[16,18,21,22,29,30]. The agonist-bound structures of ELIC in different nanodiscs also vary in the extent of these conformational changes. Comparing the ECD of unliganded and agonist-bound structures in MSP1E3D1 and spMSP1D1 nanodiscs, MSP1E3D1_ELIC shows less rotation of the ECD (0.3°) compared to spMSP1D1_ELIC (1.5°) (Fig. 3d). There is also slight variation in the size of the agonist binding site between agonist-bound structures in the order: SMA_ELIC > saposin_ELIC-MSP1E3D1_ELIC-spNW15_ELIC > spMSP1D1_ELIC-MSP1E3D1_ELIC5 (Fig. 3b). The position of the β1-β2 and β8-β9 loops are similar in all agonist-bound structures except SMA_ELIC, which more closely resembles the unliganded structures (Fig. 3c). Therefore, nanodiscs have long-range effects on ECD structure. The differences in the ECD between nanodiscs show a similar pattern to the differences in the TMD: with some variation, SMA_ELIC and MSP1E3D1_ELIC show more limited agonist-induced changes compared to spMSP1D1_ELIC and spNW15_ELIC.

## MD simulations of ELIC in different size nanodiscs

What determines the structural differences of ELIC in different nanodiscs is unclear. There is a correlation between the size of the nanodisc and the degree of agonist-dependent changes in ELIC, with larger diameter nanodiscs producing greater changes. Thus, we explored the influence of nanodisc size on ELIC structure by performing molecular dynamics (MD) simulations (triplicate 500 ns trajectories) of spMP1D1_ELIC in 9 and 11 nm MSP nanodiscs (Fig. 4a, Supplementary Fig. 8, Supplementary Table 2) as well as a planar bilayer. The

conformations of ELIC sampled in each condition differ, especially when examining the top of M2 and M4. In the 9 nm nanodisc, we observed local fluctuation around the starting spMP1D1_ELIC structure with a small increase in M4-pore axis distance in a subset of the population, and no change in the M2-pore axis distance (Fig. 4c). In contrast, the 11 nm nanodisc showed a greater increase in both M2- and M4-ion pore distance (Fig. 4d). These data suggest that the smaller 9 nm nanodisc is forcing compaction of the TMD. Interestingly, the bilayer simulation shows two locally-stable conformations: one with an increase in M4-pore axis distance only and one with the largest increase in M2-pore axis distance along with an increase in M4-pore axis distance (Fig. 4e). The top of M2 is quite dynamic throughout the simulations such that the differences between conditions are not as well appreciated from averaged plots of the M2 Cα-atom distance to the pore axis (Fig. 4b, Supplementary Fig. 9). We also examined the tilt angle of M4 relative to the pore axis (i.e., axis normal to the bilayer) (Fig. 4f). The tilt angles sampled by M4 are generally higher in the 11 nm nanodisc and planar bilayer compared to the 9 nm nanodisc. Overall, the results of the MD simulations demonstrate that ELIC structure is altered by inclusion in a nanodisc. Despite the limited time scale of these simulations, local perturbations to the ELIC TMD are observed. The 9 nm nanodisc restricts the outward movement and tilting of the top of M2 and M4. These findings generally agree with the observed effects of different nanodiscs on ELIC M2 and M4 structure by cryo-EM. The impact of the larger 11 nm nanodisc on ELIC structure is more similar to but not identical to the planar bilayer.

Two possible explanations for the difference in ELIC TMD conformation observed in our nanodisc systems include alterations to the

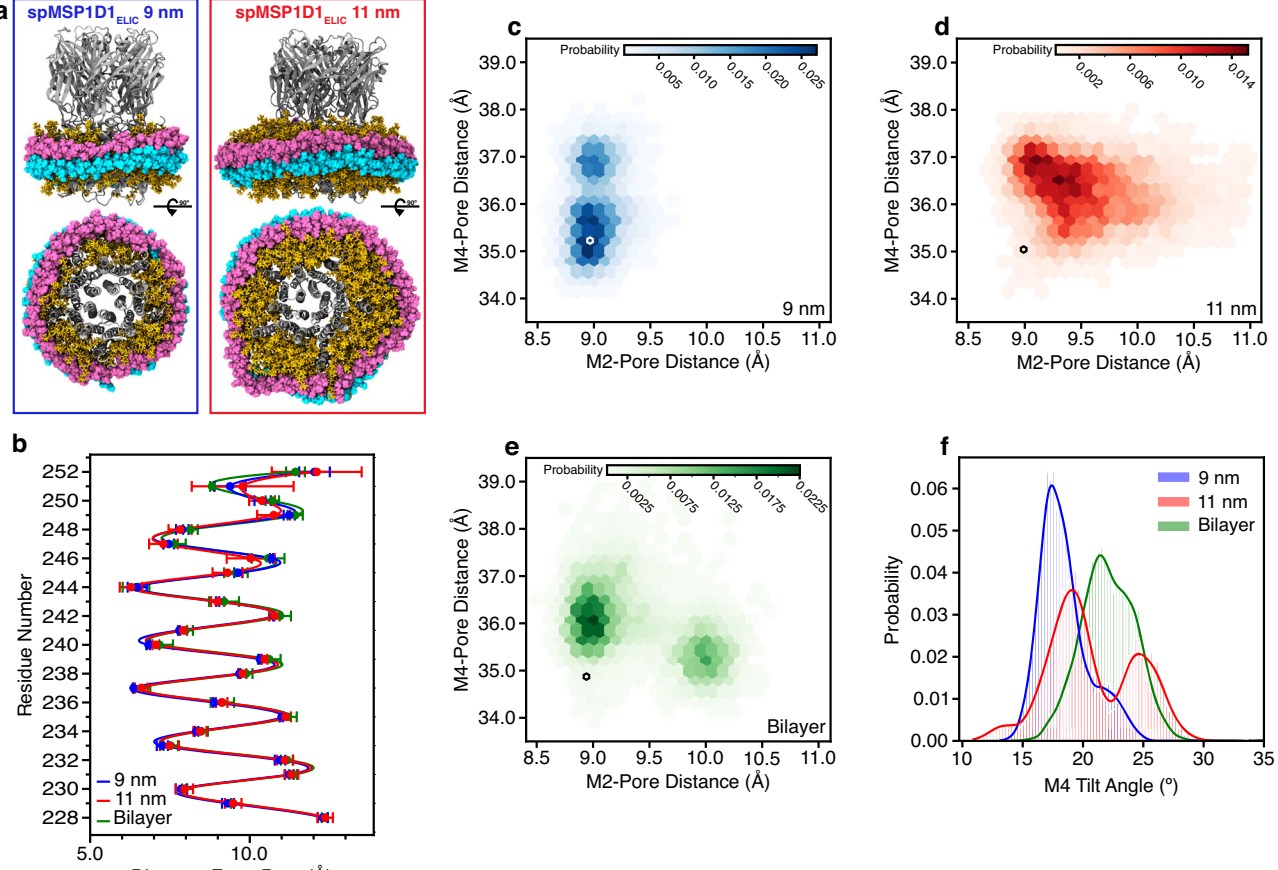

**Fig. 4 | Structure of ELIC in nanodiscs from MD simulations. a** Atomistic models of spMSP1D1$_{ELIC}$ (gray) equilibrated in a 9 nm and 11 nm MSP nanodisc. The MSP scaffolds are cyan and pink, and POPC is tan. For the top view, the ECD was removed to improve clarity of the TMD and MSP. **b** Plot of the M2 Cα-atom distance to the pore axis averaged across three independent simulations (± SD) for the 9 nm nanodisc (blue), 11 nm nanodisc (red), and bilayer (green) simulations. Two-dimensional plots of M2 (T249)- pore axis distance versus M4 (V316)- pore axis distance are shown for the 9 nm (**c**), 11 nm (**d**), and planar bilayer (**e**) simulations. The distance for the starting structure (spMSP1D1$_{ELIC}$) is shown as an empty hexagon. **f** M4 tilt angle relative to the pore axis measured above residue P405. Data is an aggregate across three independent simulations.

bilayer physical properties and direct interactions between ELIC and the MSP scaffold. Empty nanodiscs are reported to have position-dependent changes in membrane thickness, being relatively thin at the nanodisc rim to minimize hydrophobic exposure and relatively thick in the nanodisc center[8,9]. We first measured the bilayer thickness as a function of distance from the ELIC surface. In the 9 nm nanodisc, both the overall membrane thickness and the thickness of the hydrophobic core are significantly reduced compared to the 11 nm nanodisc and the planar bilayer, which show similar membrane thickness profiles near ELIC (Fig. 5a, b, Supplementary Fig. 10).

In addition to changes in membrane thickness, we also observed direct interaction between ELIC and the MSP. In the 9 nm nanodisc simulation, there is near-constant interaction between the outer face of M4 and the MSP as well as some interaction between the MSP and the β6-β7 loop (Fig. 5c). The membrane-facing residues in the middle of M4 are hydrophobic (L306-I317), and these form frequent non-polar interactions with the MSP in keeping with the interior of the scaffold displaying hydrophobic residues (Fig. 5e). However, these M4 residues, especially V314-I317, also have frequent interactions with polar and charged residues, which would be expected to be highly perturbing to the local structure. In contrast, the number of interaction sites and the frequency of interaction between M4 and the MSP is significantly reduced in the 11 nm nanodisc, confined mostly to the C-terminal residues of M4 (Fig. 5d, f). These C-terminal interactions may explain why the M4 tilt angle shows a bimodal distribution in the 11 nm nanodiscs with the larger tilt angle resulting from these

interactions pulling M4 outward (Fig. 4f). Therefore, direct interactions of M4 with the MSP scaffold could also be altering the conformation of ELIC.

## Discussion

Structures of ELIC in different nanodiscs broadly segregate into two groups. With some variability, there are those that show limited (SMA and MSP1E3D1) or greater activation (spNW15 and spMSP1D1). At saturating concentrations of agonist, ELIC is expected to be mostly desensitized at steady state[23,31]. The pore of spNW15$_{ELIC}$ shows widening at the top and narrowing at the bottom to a greater extent than any other WT agonist-bound structure. These changes are similar to desensitized conformations described for other pLGICs[16,18,29,32], suggesting that spNW15$_{ELIC}$ could be a desensitized conformation of ELIC. However, the varied effects of different nanodiscs on ELIC structure by cryo-EM raises the unsettling question of whether any of these structures are sampled in a native membrane environment. Given the apparent dependence of ELIC structure on nanodisc size, circularized nanodiscs like spNW15 or larger may be better membrane mimetics for cryo-EM studies of ELIC and other pLGICs. MD simulations combined with computational electrophysiology may also be useful to identify functionally-relevant conformations of pLGICs as was previously demonstrated for the GlyR[33,34].

This study raises the concern that structures of other pLGICs are also affected by nanodiscs. For example, the diameter of the nanodisc density of the GABA$_A$R in MSP2N2 nanodiscs was estimated to be

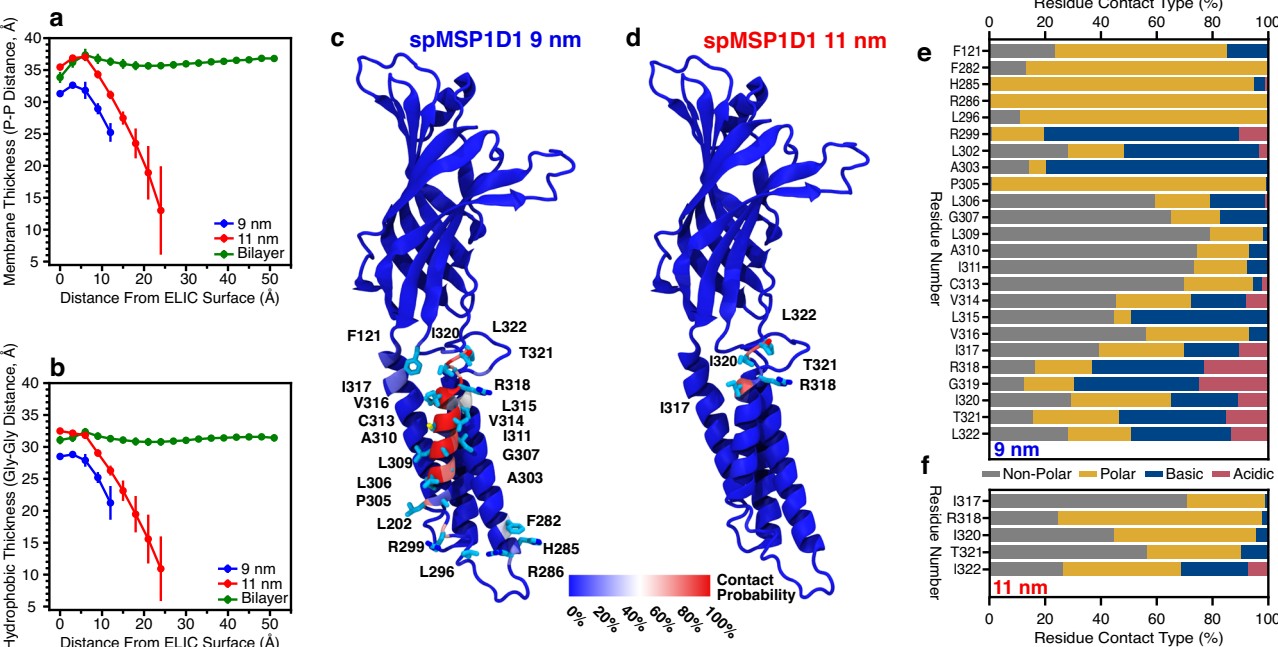

**Fig. 5 | Quantifying the effects of the nanodisc environment.** Total (**a**) and hydrophobic (**b**) bilayer thickness are displayed relative to distance from the ELIC TMD surface for the 9 nm nanodisc (blue), 11 nm nanodisc (red), and planar bilayer (green) simulations. Data is presented as average across three independent simulations (± SD). The probability of ELIC contacting the MSP is shown for spMSP1D1$_{ELIC}$ in the 9 nm nanodisc (**c**) and 11 nm nanodisc (**d**). The probability was calculated over the last 250 ns of the simulation with a contact denoted by any atom of a given ELIC residue within 4.5 Å of any atom in the MSP. Residues with low contact probability are shown in blue and high contact probability in red. Residues with >25% contact probability are explicitly shown as licorice representations. Most residues with high contact probability are in M4. For each residue with a contact probability >25%, the ratio of interacting residue types from the scaffold protein is shown for the 9 nm (**e**) and 11 nm (**f**) nanodisc simulations.

~9 nm[19] such that the GABA$_A$R is also likely to interact with the nanodisc scaffold through M4. This may explain why a recent structure of the α1β3 GABA$_A$R showed a closed activation gate despite having an estimated peak open probability of ~0.6 in HEK293 cells[35]. Structural differences were also noted between α1β3γ2 and α1β2γ2 GABA$_A$R structures in MSP2N2 and saposin nanodiscs, respectively[19,32], although these structures also differed with regards to the presence of an intact intracellular loop. In another study, a structure of agonist-bound GlyR in MSP2N2 produced a desensitized conformation, while agonist-bound GlyR in SMA produced a mixture of pre-active, open-channel and desensitized conformations[16]. While we did not identify multiple conformations of ELIC from a single dataset, we also found that SMA limits activation of ELIC. Therefore, the effects of different nanodiscs on ELIC may be present to varying degrees in other pLGICs such as the GABA$_A$R and GlyR.

In conclusion, nanodiscs affect the structure of ELIC and this effect likely relates, in part, to nanodisc size. The impact of the nanodisc may relate to direct interactions between the nanodisc scaffold and protein[36]. Therefore, the composition of the nanodisc scaffold (e.g., peptide versus synthetic polymer), irrespective of nanodisc size, may also impact protein structure. We also cannot rule out the possibility that the structural changes observed in ELIC are due to alterations in lipid composition in different nanodiscs[37-39]. It will be critical to consider the effect of nanodiscs when studying the structure of pLGICs and possibly other membrane proteins. One idea that will require further testing is that large circularized nanodiscs are better membrane mimetics.

## Methods

### Purification and reconstitution of ELIC in nanodiscs

ELIC was expressed and purified[40] using pET-26-MBP-ELIC provided by Raimund Dutzler (Addgene plasmid #39239). ELIC was expressed in OverExpress C43 (DE3) *E. coli* (Lucigen 60446-1) with Terrific Broth (Sigma T0918) using 0.1 mM IPTG (Sigma I6758) for induction. The cells were lysed using an Avestin C5 emulsifier, isolated membranes were solubilized with 1% DDM (Anatrace D310S) and purified using amylose resin (New England Biolabs E8022L). The protein was eluted with buffer A (10 mM Tris pH 7.5, 100 mM NaCl) plus 0.02% DDM, 0.05 mM TCEP (ThermoFisher Scientific T2556) and 40 mM maltose (Sigma M5885), digested overnight with HRV-3c protease (ThermoFisher Scientific 88947), and purified over a Sephadex 200 Increase 10/300 size exclusion column (Cytiva 28-9909-44) in buffer A with 0.02% DDM.

Reconstitution of ELIC in saposin, spMSP1D1 and spNW15 was performed using a liposome destabilization technique[23]. A 2:1:1 molar mixture of POPC:POPE:POPG (Avanti Polar Lipids) in chloroform was dried overnight in a desiccator, rehydrated in buffer A to 7.5 mg/ml (~10 mM), freeze-thawed 3× and extruded with a 400 nm filter (Avanti Polar Lipids). These liposomes were destabilized with DDM at a final concentration of ~0.4% DDM at RT for 3 h. Next, ELIC and the nanodisc scaffold protein was added at the following ELIC:scaffold:phospholipid molar ratios: 1:30:230 for saposin, 1:2:290 for spMSP1D1, 1:2:400 for spNW15$_{ELIC}$, and 1:2:360 for spNW15$_{ELIC5}$. This mixture yielded a final DDM concentration of ~0.2% and was rotated RT for 1.5 h, followed by Biobeads SM-2 Resin (Bio-Rad 1528920) for the removal of DDM overnight at 4 °C. The nanodisc sample was finally purified over a Sephadex 200 Increase 10/300 column in 10 mM HEPES pH 7.5 with 100 mM NaCl. Propylamine (Sigma 240958) was added to a final concentration of 50 mM at least 30 min prior to freezing for cryo-EM. These nanodisc samples were concentrated to 0.6–1.2 mg/ml. The His-tagged saposin construct was obtained from Salipro Biotech AB and purified using a Ni-NTA column (Thermo Scientific 88222). After removal of the His-tag using TEV digestion, the protein was purified by size exclusion chromatography. spMSP1D1 (Addgene plasmid #173482) and spNW15 (Addgene plasmid #173483) were gifts from Huan Bao and purified using Ni-NTA without size exclusion chromatography[3].

Reconstitution of ELIC in SMA was performed by first generating ELIC proteoliposomes. 2:1:1 POPC:POPE:POPG was solubilized in buffer A with ~40 mM CHAPS (Anatrace C316S), after which ELIC was added (100 µg per mg of lipid) for 30 min at RT. Next BioBeads was added to remove DDM rotating for ~2.5 h. The proteliposome suspension was extruded with a 100 nm filter (Avanti Polar Lipids). To form SMA nanodiscs, 20% SMALP 300 (Orbiscope) was added to the Biobead-free proteoliposomes to a final concentration of 2.5%, and agitated at RT for 2 h in the absence of light. To isolate ELIC SMA nanodiscs from empty SMA nanodiscs, the sample was purified over a Ni-NTA column (WT ELIC binds to Ni-NTA likely through native histidine residues) and eluted with 30 mM imidazole (Sigma I5513). The eluate was then purified over a Sephadex 200 Increase 10/300 column in 10 mM HEPES pH 7.5 with 100 mM NaCl. Fractions containing the nanodisc sample were concentrated to ~1.2 mg/ml and propylamine was added to 50 mM for cryo-EM imaging.

## Cryo-EM sample preparation and imaging

3 µl of ELIC in nanodiscs were pipetted onto Quantifoil R2/2 copper grids (which had previously been cleaned using a Gatan Solaris 950 in a $H_2/O_2$ plasma for 60 s), after which each grid was blotted for 2 s in a 100% humidity environment and vitrified in liquid ethane using a Vitrobot Mark IV (ThermoFisher Scientific). Each grid was then imaged on a Titan Krios 300 kV Cryo-EM equipped with a Falcon 4 Direct Electron Detector (ThermoFisher Scientific), except the spNW15$_{ELIC5}$ dataset which was imaged on a Glacios 200 kV Cryo-EM equipped with a Falcon 4 Direct Electron Detector. Single particle cryo-EM data was acquired using counting mode on the Falcon 4 with the EPU software (version 2.12.1.2782 and 3.1.0.4506REL). Movies were collected using a pixel size of 0.842 Å for SMA$_{ELIC}$, 0.657 Å for saposin$_{ELIC}$, spMSP1D1$_{ELIC}$ and apo-spMSP1D1$_{ELIC}$, and 1.081 Å for spNW15$_{ELIC}$, with a defocus range of −0.8 to −2.4 µm. For spNW15$_{ELIC5}$, movies were collected using a pixel size of 1.184 Å along with defocus range of −1 to −2.4 µm. For SMA$_{ELIC}$, each movie consisted of 46 individual frames with a per-frame exposure time of 200 ms, resulting in a dose of 49.45 electrons per Å². For saposin$_{ELIC}$, spMSP1D1$_{ELIC}$ and apo-spMSP1D1$_{ELIC}$, each movie consisted of 49 individual frames with a total dose of 47.55, 46.93 and 51.54 electrons per Å², respectively. For spNW15$_{ELIC}$ and spNW15$_{ELIC5}$, each movie consisted of 50 and 45 individual frames with a total dose of 54.6 and 50.09 electrons per Å², respectively.

## Single particle analysis and model building

Single particle cryo-EM datasets of spNW15$_{ELIC}$ and spNW15$_{ELIC5}$ were processed in Relion 3.1[41] and CryoSPARC 3.3.2[42]. All other cryo-EM datasets were processed in Relion 3.1[41]. Similar processing was followed for all datasets in Relion 3.1[41]. Movies were motion corrected with MotionCor2[43], and contrast transfer function (CTF) determined with GCTF v1.06[44]. Particles were initially picked using LoG-based autopicking, followed by 2D class averaging to generate 2D classes for template-based picking. Particles were extracted and subjected to multiple rounds of 2D and 3D classification using a mask diameter of 140 Å. The initial model for 3D classification was generated using a 40 Å low-pass filtered map of MSP1E3D1$_{ELIC}$ (EMD-27218), and 3D classifications and subsequent 3D refinements were performed using C5 symmetry. 3D classification including focused classification of the transmembrane domain (TMD) produced only a single conformation from each dataset. The best 3D refine map was then subject to post-processing, CTF refinement and Bayesian polishing. The spNW15$_{ELIC}$ and spNW15$_{ELIC5}$ datasets were processed similarly in CryoSPARC 3.3.2[42], where movie frames were aligned with patch motion correction and contrast transfer function (CTF) estimation was done using patch CTF estimation. For example, for spNW15$_{ELIC}$, 2,524,299 particles were picked using templates generated from MSP1E3D1$_{ELIC}$ (EMD-27218) and were subjected to several rounds of 2D classification to remove junk particles. An ab initio model was generated and refined with several rounds of heterogeneous refinements, followed by a reconstruction of the final map using non-uniform refinement (C5 symmetry)[42]. Final maps obtained from Relion 3.1 and CryoSPARC 3.3.2 for spNW15$_{ELIC}$ and spNW15$_{ELIC5}$ were identical, although spNW15$_{ELIC}$ yielded higher resolution in Relion 3.1[41] and spNW15$_{ELIC5}$ in CryoSPARC 3.3.2[42].

An initial model of ELIC was obtained with MSP1E3D1$_{ELIC}$ (PDB 8D66 [https://doi.org/10.2210/pdb8D66/pdb]), which was used to perform real space refinement in PHENIX 1.19.2[45]. The structure was then manually built into the cryo-EM density map using COOT 0.9.6[46] followed by iterative real space refinement in PHENIX and manual adjustments in COOT. Propylamine was fit in the density in the agonist binding site based on the predicted orientation of the amine group in cysteamine (another agonist of ELIC) from MD simulations[24]. To estimate the diameter of the nanodisc, unsharpened maps were low-pass filtered (8 Å) using relion image handler, as done by Noviello et al.[29]. The pointer atoms were placed at the edges and center of the map set at a contour of 1 σ and distances were measured using the distance tool in COOT. 3D volume visualization and molecular image preparation were performed using PyMOL 2.5.2 and ChimeraX 1.6.1.

## Modeling ELIC in a nanodisc

To further examine the effect of nanodisc diameter on the conformation and dynamics of ELIC, we used molecular dynamics (MD) to simulate spMSP1D1$_{ELIC}$ in two nanodiscs of differing size (9 nm and 11 nm), with an all-atom model chosen to observe relative domain motions of the transmembrane helices. POPC was chosen as the sole lipid for these systems to avoid the uncertainty of placing different lipids in the model nanodisc, since significant lipid diffusion is not expected during the time scale of the simulation. By using the same starting structure of ELIC in both nanodisc systems, we can examine the effect of nanodisc size on ELIC structure. Starting with spMSP1D1$_{ELIC}$, we modeled the missing residues (R318, G319, I320, T321, L322) assuming M4 remains an α-helix through its C-terminus. ELIC agonist, propylamine, was modeled into its binding site as in the cryo-EM structure. Parameters for propylamine were generated using the CGenFF server[47,48]. The local pKa of ionizable groups in ELIC side chains was determined with PROPKA3; this resulted in no protonation/deprotonation of any side chains in the protein with the system pH at 7.0. The N-terminus P11 was acetylated and the C-terminus was left charged as this is the true terminus. This structure was imported into CHARMM-GUI[49] and placed in either an MSP1D1-33 (9 nm) nanodisc, MSP1E2D1 (11 nm) nanodisc, or planar bilayer[8]. The PPM server[50] was used to orient ELIC such that the ion conduction pathway aligned to the normal of the plane delineated by the MSP bundle or membrane normal in the case of the planar bilayer system. The nanodisc and bilayer were composed of only POPC lipids to eliminate local lipid composition as a factor in the conformational dynamics of ELIC. The ELIC-nanodisc construct was then solvated with the TIP3 water model[51] to provide ~2 nm of buffer between protein and the simulation box edge. Each system was then ionized with 150 mM NaCl and neutralized. The final simulation box measured $15.5 \times 15.5 \times 16.4$ nm³ with 363,774 atoms for the 9 nm nanodisc simulation system, $17.0 \times 17.0 \times 17.1$ nm³ with 457,197 atoms for the 11 nm nanodisc simulation system, and $12.1 \times 12.0 \times 14.5$ nm³ with 162,265 atoms for the planar bilayer system.

Each simulation system was equilibrated for 50 ns. In each case, the backbone of all protein segments and the heavy atoms of phospholipids were harmonically restrained to their initial coordinates ($k = 500$ kcal/mol/nm²) for 5 ns. The harmonic position restraints were slowly removed over 20 ns with reduction of the harmonic force constant by 50 kcal/mol/nm² every 2 ns. The system was then allowed 25 ns of equilibration without restraint, followed by 500 ns of unrestrained MD, which has been sufficient to capture rigid-body motion of ELIC transmembrane helices[28]. Simulations were carried out in an NPT ensemble. Pressure was maintained at 1 atm using the Nosé-Hoover

Langevin piston method[52,53] with a piston period of 100 fs and piston decay of 50 fs. Temperature was maintained at 310 K with Langevin dynamics and a damping coefficient of 1 ps⁻¹. A 2 fs timestep was used for integration. Short-range non-bonded interactions were cutoff after 12.0 Å with a switching function applied after 10.0 Å. Long-range electrostatics were handled using the particle mesh Ewald sums method[53]. MD simulation was carried out with NAMD 2.14[54]. VMD (version 1.9.4.a55)[55] was used for molecular visualization and quantitative analysis, along with in-house code (Python 3.6.8) available at Zenodo (10.5281/zenodo.10214906)[56] and Dryad (10.5061/dryad.z8w9ghxk5)[57]. The CHARMM36[58] parameter set was used for lipids and ions with CHARMM36m[59] and cation-π corrections[60] being applied to protein segments.

**Analysis of ELIC in nanodisc and bilayer systems.** It is known that nanodiscs do not retain a perfect circular shape but demonstrate multiple conformational clusters that are ellipsoid[9,11]. Therefore, to demonstrate stability of nanodisc size in the 9 nm and 11 nm simulation systems, we generated an ellipse of best fit to the backbone atoms in both MSPs[9]. The ellipse of best fit was recorded every 200 ps. The mean and standard deviation of nanodisc diameter was calculated over the last 250 ns of the simulation.

To assess the effect of nanodisc size on the local membrane environment, we measured membrane thickness as a function of distance from the protein as well as two-dimensional thickness as a function of distance from the ion conduction pore; similar measurements have been made previously in empty nanodiscs[8,9]. Each frame of the simulation trajectory was aligned using the TMD of ELIC (residues 201–322). The instantaneous membrane midplane was taken to be the z-component of the geometric center of all phosphorous atoms in the system with lipids marked as being "upper" or "lower" leaflet depending on their position relative to the membrane midplane. All lipids in the system were then placed in a bin corresponding to the smallest distance between the phosphorous atom of each lipid and any TMD backbone atom. Bins were 0.3 nm wide and spanned from 0 nm to the radial width of the nanodisc (i.e., 4.5 nm for the 9 nm nanodisc system and 5.5 nm for the 11 nm nanodisc system). The instantaneous height of the bin was taken as the average height of all lipids within a bin, calculated using either the phosphorous atom or the geometric center of the glycerol backbone. The membrane thickness was then calculated as the distance between the height of upper lipids and lower lipids in each bin. The membrane thickness was measured every 200 ps over the last 250 ns of the simulation. For two-dimensional membrane thickness measurement, a similar process was used as above except binning was performed using polar coordinates (r,q) relative to the ion conduction pore with radial and theta bins at 5 Å and p/15 radian intervals, respectively. Again, membrane thickness was measured every 200 ps over the last 250 ns of the simulation.

### Reporting summary

Further information on research design is available in the Nature Portfolio Reporting Summary linked to this article.

## Data availability

The data supporting the findings of this study are available within the paper and supplementary information files. The cryo-EM maps have been deposited in the Electron Microscopy Data Bank (EMDB) under accession codes EMD-28829 for SMA_ELIC, EMD-28830 for saposin_ELIC, EMD-28831 for spMSP1D1_ELIC, EMD-28832 for apo-spMSP1D1_ELIC, EMD-41673 for spNW15_ELIC, and EMD-41672 for spNW15_ELIC5. The structural coordinates have been deposited in the RCSB Protein Data Bank (PDB) under the accession codes 8F32 for SMA_ELIC, 8F33 for saposin_ELIC, 8F34 for spMSP1D1_ELIC, 8F35 for apo-spMSP1D1_ELIC, 8TWZ for spNW15_ELIC, and 8TWV for spNW15_ELIC5. The MD simulation data from reduced trajectories are deposited in Dryad (https://doi.org/10.5061/dryad.z8w9ghxk5). Source data for all figures are also provided as a Source Data file. Source data are provided with this paper.

## Code availability

In-house analysis scripts for the MD simulation data are available in online repositories, Zenodo (https://doi.org/10.5281/zenodo.10214906) and Dryad (https://doi.org/10.5061/dryad.z8w9ghxk5).

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

## Acknowledgements

This study was supported by grants R35GM137957 to WC, F32GM139351 to JP, NSF2152059 to GB, and the Foundation for Anesthesia Education and Research Mentored Research Training Grant to MJA. We are grateful to Dr. Joe Henry Steinbach and Dr. Alex Evers for helpful discussions.

## Author contributions

W.W.C., J.T.P., V.D., M.J.A. and G.B. conceived the project and designed the experimental procedures. N.M.D. carried out the protein expression and purification. V.D., J.T.P. and B.K.T. prepared the nanodisc samples for cryo-EM and performed the single particle analysis. M.J.A. performed the MD simulations and analysis of the MD trajectories. M.R. and J.A.J.F. performed grid preparation and image acquisition for cryo-EM. All authors reviewed the paper.

## Competing interests

The authors declare no competing interests.
