## [Peer Review File · Nature Communications]

Reviewers' Comments:

Reviewer #1:

Remarks to the Author:

Remarks to the Author

The manuscript by Dalal et al. delves into the thorny topic of studying membrane mimics of pentameric ligand gated ion channels. This is an important topic because of the general overall interest in pLGICs and the well known, yet often avoided, effects that membrane mimics have on channel conformation and function. More generally, the authors are to be commended for taking on this study given the overall significance of nanodiscs and amphiphilic polymers in the study of membrane proteins.

The authors carried out a systematic investigation of the model pLGIC, ELIC, using a combination of single particle cryo-EM and MD simulations, together with several different membrane mimics that include nanodiscs and amphipathic polymers, with the overall message being that a larger nanodisc is a more faithful mimic of the membrane yet why it is, and especially why smaller nanodiscs are not as faithful, is still not understood. While the work is overall satisfactory, the writing could be substantially improved and, in multiple places, especially in the conclusion, the manuscript could be tightened up and shortened.

Comments

1. There may be some dissension on this point in the field, but my general understanding is that "Nanodisc" or "nanodisc" are used to refer to a MSP and lipid complex whereas SMA, amphipols and saposin are referred to as amphipathic polymers. See for example: <https://doi.org/10.1002/pro.3994>
2. Lines 164-170. As the authors note, the densities for M4 helices (Supp Fig 7) are weak and it is challenging to see from the figure how one could reliably fit the main chain. For example, in Supp Fig 7b of the saposin complex, most of the main chain atoms are not in density. If there is not density for main chain atoms, then it is not appropriate to model these elements of structure, at least using the single particle cryo EM density map(s) as guides.
3. Figure 3. The colors chosen for the 5 different structures are tough to separate, with saposin (dark grey) much like 'sp' (light grey) and so on. The authors should consider using an established color palette to help them chose a spectrum of colors that are most easy for the reader to distinguish (see for example Fig 1 in <https://doi.org/10.1002/rth2.12308>).
4. In Figure 3d, it is difficult to visualize the extent to which W206 occupies either a membrane-facing conformation or an inter subunit space. The authors should either chose a different view, that places the conformational difference in the plane of the page, or should use a stereo figure.
5. Lines 189-198. Again, the authors cannot discern motion, movement or order of motion from the analysis of static structures. Rather, they can state which structures adopt specific conformations and what the conformational differences are between those states/structures.
6. Figure 5. Because the authors emphasize residue L315 they should label the residue in Supp Figure 7. If L315, however, is the leucine residue clearly visible in Supp Figure 7b, however, then the reliability of the residue is called into question, due to the fact that density for the residue is either weak or nonexistent.
7. Lines 249 onward. I'll leave it to the computational reviewer(s), but must ask whether 500 ns is a sufficiently long time for relatively large-scale conformational changes to occur and whether the authors tested the extent that the systems had reached global minimums by starting at different conformations.
8. Line 314. The statement that "...the MD simulations of spMSP1D1ELIC suggest that this structure is not very stable..." is a bit of an oxymoron. Of course the structure (as the complex

with nanodisc) must be stable. The more precise question is whether, when outside of the nanodisc and in a membrane bilayer, the structure is stable. As this is likely what the authors mean to communicate, they should revise the respective sentences appropriately.

9. Line 353. Again, the authors cannot determine, from the cryo-EM data, whether 'differences in the structures' are concerted.

10. Line 358. LGICs are allosteric machines. It is no surprise that there is coupling between the TMD and ECD.

Minor comments

1. In the Abstract, I'm not sure how the authors can know that the "...nanodiscs scaffolds produce concerted conformational changes...". The changes could be sequential and still end up yielding the same final structures. The use of 'nanodiscs' is also either grammatically incorrect or simply awkward and unconventional.

2. Related to the single particle analysis, I'm a bit surprised that the authors did not find more than a single 3D class for each of the receptor complexes and, related to this point, it is not clear how thoroughly they carried out 3D classification, especially focused classification on the TMD region.

3. Line 126. Weaker density for F247 does not mean a more dynamic structure as the residue could also simply occupy several different conformations, statically. More accurate to state that the residue exhibits conformational heterogeneity, which could be dynamic or static in nature.

4. Line 260. 'Since' is time, 'Because' is reason, i.e. it should read "Because nanodiscs...".

5. Fig. 2d and e and supplementary fig. 6c are redundant. Figures Fig. 2e and supplementary fig. 6c could be combined.

6. The structure alignments in supplementary fig. 6a and b are not presented along with the structural comparison in the appropriate context (line 147-160) and could be modified to better accommodate readers.

Reviewer #2:

Remarks to the Author:

solubilize the transmembrane portions of the protein for use in cryo-EM studies. However, in recent years, there has been increasing concern that the nature and size of the disc may possibly give rise to artefactual and misleading results, especially for membrane proteins like ligand-gated ion channels, which are known to be especially sensitive to the lipid environment. In this work Dalal et al, approach this problem systematically by studying the well-characterised pLGIC ELIC in a series of different nanodiscs. The results confirm the notion that the size of nanodisc can indeed strongly influence the observed structure. The authors go on to investigate possible reasons for this via the use of some (limited) MD simulations. As well as the possibility of direct interaction between the channel protein and the edge of the nanodisc, the lipid thickness is also touted as a possible explanation for the changes in observed structure.

Overall, the work is systematic and performed well. The manuscript is clearly written if a little verbose and repetitive in places. I think the following points would need addressing before being considered for publication:

Whilst the MD simulations look sound and are readily interpretable within the context of the EM data, they are very short (500 ns) single simulations. Thus, the differences particularly between the bilayer and the nanodisc simulations could just be chance. Repeats (preferably more than 1 repeat) for each system are required to put the interpretations on a more sound footing.

The authors quantify the time/% of interactions between the protein and nanodiscs but it would be good to know more about the nature of the interactions – are they mostly hydrogen bonds for example?

It would be beneficial to expand the Discussion section a little to talk about similar issues that were observed with the Glycine receptor – especially Du et al's original open structure from 2015 which many research groups quickly realised was likely to be artefactual and indeed led to further discussions about the nature of the state (see Cerdan et al <https://pubmed.ncbi.nlm.nih.gov/30220542/> and also Damgen et al <https://pubmed.ncbi.nlm.nih.gov/31753620/>).

The authors provide EM models, but there is no mention of the MD data. The input and parameter files at the very least should be provided and preferably some (skipped) trajectories.

Minor comments

The introduction around p61 – there are some LGIC studies in nanodisc studies missing – see for example. Kumar et al 2022 who used asolectin nanodiscs with the E3D1 membrane scaffolding protein (PMID: 35982060. DOI: 10.1038/s41467-022-32594-5)

Also the effects of lipid nanodisc composition has been investigated by others and much of this is not cited very well either (see <https://pubs.acs.org/doi/10.1021/acs.analchem.2c01488> for example).

P89 – the spytag technology should be referenced properly.

P122 – would be useful to be reminded of the diameter.

P293 – surely that profile is consistent with open state rather than desensitized?

Is the final recommendation simply to use the biggest nanodisc one has access to?

Introductory Comments

We are grateful for your consideration of our manuscript, and are pleased to present this revision. It is important to note that during the revision process, we discovered a critical error in the structural analysis. Specifically, the pixel size provided by our EM facility was incorrect leading to an improper scaling that affected all the structures. Therefore, we needed to re-analyze all the structures using the correct pixel sizes and re-run all the MD simulations using the corrected structures. The original manuscript described some differences between the new structures and previously published structures of ELIC (e.g. ECD expansion) that were simply an artifact of the incorrect pixel size. This has now been removed from the results. Since our new structures show excellent agreement with previously published structures in regions where minimal change is expected (e.g. the ECD), we are now confident that the well-calibrated pixel sizes are yielding correct structures. Overall, the general findings and conclusions of the study have not changed (i.e. nanodiscs affect ELIC structure), even though the specific results are somewhat different. We are very pleased that this error was discovered prior to publication, so that we can provide a rectified manuscript for your review.

During the revision process, we also decided to add two new structures. These are WT ELIC and the ELIC5 mutant reconstituted in the larger circularized nanodisc, spNW15 (spNW15_{ELIC} and spNW15_{ELIC5}). spNW15 is the largest of all the nanodiscs analyzed, and the changes observed in the WT structure support the notion that nanodisc size impacts ELIC structure. We believe this addition enhances the quality of the study and strengthens its conclusion.

A critique of both reviewers was that the original manuscript was too long and repetitive. We have significantly revised the writing to make it more concise, and hope the reviewers will find it easier to read and review.

Below are specific responses to the reviewer's comments which we have addressed in the manuscript.

REVIEWER COMMENTS

Reviewer #1 (Remarks to the Author):

Remarks to the Author

The manuscript by Dalal et al. delves into the thorny topic of studying membrane mimics of pentameric ligand gated ion channels. This is an important topic because of the general overall interest in pLGICs and the well known, yet often avoided, effects that membrane mimics have on channel conformation and function. More generally, the authors are to be commended for taking on this study given the overall significance of nanodiscs and amphiphilic polymers in the study of membrane proteins.

The authors carried out a systematic investigation of the model pLGIC, ELIC, using a combination of single particle cryo-EM and MD simulations, together with several different membrane mimics that include nanodiscs and amphiphilic polymers, with the overall message being that a larger nanodisc is a more faithful mimic of the membrane yet why it is, and especially why smaller nanodiscs are not as faithful, is still not understood. While the work is overall satisfactory, the writing could be substantially improved and, in multiple places, especially in the conclusion, the manuscript could be tightened up and shortened.

We agree that the original manuscript was too long and repetitive; the revised manuscript is much more succinct.

Comments

1. There may be some dissension on this point in the field, but my general understanding is that “Nanodisc” or “nanodisc” are used to refer to a MSP and lipid complex whereas SMA, amphipols and saposin are referred to as amphipathic polymers. See for example: <https://doi.org/10.1002/pro.3994>

We now introduce each nanodisc with more explicit definition of the terms. For SMA, we introduce it as a “polymer nanodisc which uses a synthetic polymer”. We introduce saposin as “saposin nanodiscs”. For the rest of the manuscript, we refer to all as “nanodiscs” with a lower-case “n” for simplicity. We distinguish this from the original **N**anodiscs, which we refer to as “MSP nanodiscs”. We are not the first to refer to SMA or saposin nanoparticles as nanodiscs (e.g. polymer or synthetic nanodiscs for SMA). Therefore, we believe this is an acceptable and easily recognizable way to refer to these particles.

2. Lines 164-170. As the authors note, the densities for M4 helices (Supp Fig 7) are weak and it is challenging to see from the figure how one could reliably fit the main chain. For example, in Supp Fig 7b of the saposin complex, most of the main chain atoms are not in density. If there is not density for main chain atoms, then it is not appropriate to model these elements of structure, at least using the single particle cryo EM density map(s) as guides.

In all the structures, the density of M4 is indeed much weaker in the sharpened maps (Figure 1). However, the density of M4, especially for the peptide backbone, is clearly appreciated in the unsharpened maps (except for spNW15_{ELIC}), and we have demonstrated this in Supplementary Figure 6. Based on these maps, we have confidence about the orientation of the M4 helix (i.e. the backbone atoms of the helix). An important finding of the study is that M4 orientation is significantly impacted by the nanodiscs (Figure 2d and Figure 4f). Because there is no density for M4 in spNW15_{ELIC}, we did not model M4 in this structure.

3. Figure 3. The colors chosen for the 5 different structures are tough to separate, with saposin (dark grey) much like ‘sp’ (light grey) and so on. The authors should consider using an established color palette to help them chose a spectrum of colors that are most easy for the reader to distinguish (see for example Fig 1 in <https://doi.org/10.1002/rth2.12308>).

We have altered the colors and made our best effort using established color palettes to pick colors that can be easily distinguished. This was done with the aid of a colleague who is color-blind. Nevertheless, some colors are still somewhat similar (e.g. SMA_{ELIC} and spNW15_{ELIC5}); this is difficult to avoid as we are now comparing nine different structures. To facilitate color differentiation, we ensured that the structures being compared in the main figures are assigned the most contrasting colors.

4. In Figure 3d, it is difficult to visualize the extent to which W206 occupies either a membrane-facing conformation or an inter subunit space. The authors should either chose a different view, that places the conformational difference in the plane of the page, or should use a stereo figure.

This image is now in Supplementary Figure 7. We believe the perspective of this image is optimal to demonstrate the change in W206 orientation. Additionally, we added a sentence in the figure legend to say that when W206 is pointing down and towards the reader in the image,

it is facing the membrane. Using a stereo view would change the colors which we want to keep consistent.

5. Lines 189-198. Again, the authors cannot discern motion, movement or order of motion from the analysis of static structures. Rather, they can state which structures adopt specific conformations and what the conformational differences are between those states/structures.

We have removed the words “move” and “movement” throughout the manuscript when describing changes in structures, and replaced it with words that simply describe the conformational differences. When we state “order”, we simply mean in order from least to greatest, not in order of motion. This should be made clear by removing the word “move”.

6. Figure 5. Because the authors emphasize residue L315 they should label the residue in Supp Figure 7. If L315, however, is the leucine residue clearly visible in Supp Figure 7b, however, then the reliability of the residue is called into question, due to the fact that density for the residue is either weak or nonexistent.

The original purpose of showing the C-alpha atom of L315 was to illustrate changes in the position of M4. The change in M4 orientation is now shown in Figure 2d using a tilt angle. We have removed description of the position of L315 as we now find it less helpful in describing the orientation of M4 in the different structures. Rather, the images in Fig. 2d of M4 and the graph of M4 tilt angle describe the structural differences sufficiently.

7. Lines 249 onward. I'll leave it to the computational reviewer(s), but must ask whether 500 ns is a sufficiently long time for relatively large-scale conformational changes to occur and whether the authors tested the extent that the systems had reached global minimums by starting at different conformations.

As requested by reviewer 2, we have performed three replicates of each of the simulations for a total of 1.5 microseconds of simulation time for each model. Each replicate was an independently generated nanodisc or bilayer system. Since the measurements such as M4 tilt or the distance of M2/M4 from the pore axis show a mostly gaussian distribution, we can be reasonably confident that the system has reached a conformational minimum, although there are clearly multiple local minimums in some cases (e.g. the bimodal distribution of M4 tilt angle in the 11 nm nanodisc). Therefore, while it is unlikely that the system has reached a global minimum, we argue that the simulations performed in this study are sufficient to support the conclusion that the nanodisc affects the conformation of the protein. What can be detected from the MD simulations are more local effects of the nanodisc on ELIC structure, as evidenced by the fact that M4 tilt is most significantly impacted by the nanodisc environment.

8. Line 314. The statement that “...the MD simulations of spMSP1D1ELIC suggest that this structure is not very stable...” is a bit of an oxymoron. Of course the structure (as the complex with nanodisc) must be stable. The more precise question is whether, when outside of the nanodisc and in a membrane bilayer, the structure is stable. As this is likely what the authors mean to communicate, they should revise the respective sentences appropriately.

We have substantially revised the discussion section, as requested, to make it more concise and readable. In the process, that statement was deleted.

9. Line 353. Again, the authors cannot determine, from the cryo-EM data, whether ‘differences in the structures’ are concerted.

We agree with the reviewer and have completely removed the word “concerted” from the manuscript.

10. Line 358. LGICs are allosteric machines. It is no surprise that there is coupling between the TMD and ECD.

Indeed, this is the case, and we have deleted this statement when editing the Discussion.

Minor comments

1. In the Abstract, I’m not sure how the authors can know that the “...nanodiscs scaffolds produce concerted conformational changes...”. The changes could be sequential and still end up yielding the same final structures. The use of ‘nanodiscs’ is also either grammatically incorrect or simply awkward and unconventional.

We have removed the word ‘concerted’. As mentioned above, we are continuing to use the term ‘nanodiscs’ to refer to the discoidal particles that can be formed using MSP, saposin and SMA.

2. Related to the single particle analysis, I’m a bit surprised that the authors did not find more than a single 3D class for each of the receptor complexes and, related to this point, it is not clear how thoroughly they carried out 3D classification, especially focused classification on the TMD region.

We have performed global 3D classification of each dataset, as well as focused 3D classification of just the TMD. In all cases, we did not identify multiple distinct classes. This is now reported in the Methods section.

3. Line 126. Weaker density for F247 does not mean a more dynamic structure as the residue could also simply occupy several different conformations, statically. More accurate to state that the residue exhibits conformational heterogeneity, which could be dynamic or static in nature.

We agree that the weaker side chain density is more accurately described as exhibiting conformational heterogeneity. We now use the terms “conformational heterogeneity” throughout the manuscript as a potential explanation for regions that show weaker density. Regarding F247, the commentary on this side chain has now been deleted. We have decided not to compare the orientation of side chains in M2 such as F247 (16’) or 2’, which generally show weaker densities. Rather, we have emphasized backbone structure by comparing the distance of M2 C-alpha atoms to the ion pore axis (Fig. 2c). We are more confident about these differences. At resolutions of 3-3.5 Å, we are observing differences of ~1-2 Å in the coordinates of M2 C-alpha atoms (e.g. S250 at the top of M2 in Fig. 2c) between structures in different nanodiscs, which are significant differences.

4. Line 260. ‘Since’ is time, ‘Because’ is reason, i.e. it should read “Because nanodiscs...”.

In the process of revision, this phrase has been removed.

5. Fig. 2d and e and supplementary fig. 6c are redundant. Figures Fig. 2e and supplementary fig. 6c could be combined.

There are now no longer duplications of graphs/images in different figures.

6. The structure alignments in supplementary fig. 6a and b are not presented along with the structural comparison in the appropriate context (line 147-160) and could be modified to better accommodate readers.

We no longer make this point in the manuscript and have removed the structural alignments.

Reviewer #2 (Remarks to the Author):

solubilize the transmembrane portions of the protein for use in cryo-EM studies. However, in recent years, there has been increasing concern that the nature and size of the disc may possibly give rise to artefactual and misleading results, especially for membrane proteins like ligand-gated ion channels, which are known to be especially sensitive to the lipid environment. In this work Dalal et al, approach this problem systematically by studying the well-characterised pLGIC ELIC in a series of different nanodiscs. The results confirm the notion that the size of nanodisc can indeed strongly influence the observed structure. The authors go on to investigate possible reasons for this via the use of some (limited) MD simulations. As well as the possibility of direct interaction between the channel protein and the edge of the nanodisc, the lipid thickness is also touted as a possible explanation for the changes in observed structure.

Overall, the work is systematic and performed well. The manuscript is clearly written if a little verbose and repetitive in places. I think the following points would need addressing before being considered for publication:

Whilst the MD simulations look sound and are readily interpretable within the context of the EM data, they are very short (500 ns) single simulations. Thus, the differences particularly between the bilayer and the nanodisc simulations could just be chance. Repeats (preferably more than 1 repeat) for each system are required to put the interpretations on a more sound footing.

Using the new corrected structures, we have repeated the MD simulations for 3 replicates at 500ns each. While there is some variability between replicates, there are reproducible differences in the conformational distribution of M4 and M2 between the various structures that indicates an effect of the nanodisc on ELIC structure.

The authors quantify the time/% of interactions between the protein and nanodiscs but it would be good to know more about the nature of the interactions – are they mostly hydrogen bonds for example?

We have now analyzed the per residue interactions of M4 with the nanodisc scaffold, categorizing the MSP interacting residue as polar, non-polar, acidic or basic (Figure 5e and 5f). This analysis shows that many of the hydrophobic residues in M4 are forming unfavorable polar interactions with MSP residues. This is especially the case in the 9 nm nanodisc, which is likely to produce a significant perturbing effect on ELIC structure.

It would be beneficial to expand the Discussion section a little to talk about similar issues that were observed with the Glycine receptor – especially Du et al's original open structure from 2015 which many research groups quickly realised was likely to be artefactual and indeed led to further discussions about the nature of the state (see Cerdan et al <https://pubmed.ncbi.nlm.nih.gov/30220542/> and also Damgen et

al <https://pubmed.ncbi.nlm.nih.gov/31753620/>).

In the first paragraph of the discussion, we state that the effects of nanodiscs on ELIC structure raises the question of whether some or all of these structures are artefactual. We then raise the point that MD simulations coupled with computational electrophysiology could be a useful approach to identify functionally-relevant conformations in a lipid membrane environment. Here, we referenced the studies of Cerdan et al and Damgen et al on the open-channel structure of the Glycine receptor.

The authors provide EM models, but there is no mention of the MD data. The input and parameter files at the very least should be provided and preferably some (skipped) trajectories.

We have uploaded input and parameter files, data from reduced trajectories and analysis code to Dryad and GitHub. This is now indicated in the Data Availability and Code Availability sections. We also added Supplementary Table 2, which summarizes the details for each simulation condition.

Minor comments

The introduction around p61 – there are some LGIC studies in nanodisc studies missing – see for example. Kumar et al 2022 who used asolectin nanodiscs with the E3D1 membrane scaffolding protein (PMID: 35982060. DOI: 10.1038/s41467-022-32594-5)

We added two references to this sentence on GlyR structures in MSP1E3D1 nanodiscs (Kumar et al 2020 and 2022).

Also the effects of lipid nanodisc composition has been investigated by others and much of this is not cited very well either (see <https://pubs.acs.org/doi/10.1021/acs.analchem.2c01488> for example).

We added a comment in the conclusion that we cannot rule out the possibility that the effects of nanodiscs on ELIC structure are a consequence of differences in lipid composition in the nanodisc, and referenced: DOI <https://doi.org/10.1038/s42003-021-01711-3>, DOI <https://doi.org/10.1021/acs.analchem.0c00786>, doi: <https://doi.org/10.1101/2023.06.02.543293>.

P89 – the spytag technology should be referenced properly.

We added this reference to this sentence.

P122 – would be useful to be reminded of the diameter.

This is no longer relevant in the revised manuscript.

P293 – surely that profile is consistent with open state rather than desensitized?

Functional annotation of the structures is challenging without performing some additional analysis like computational electrophysiology. In the revised manuscript, we have been reticent to make strong statements regarding functional annotation, except for some comments that spNW15_{ELIC} could represent a desensitized conformation with 9' forming a hydrophobic gate.

Is the final recommendation simply to use the biggest nanodisc one has access to?

The agonist-bound WT ELIC structures in larger nanodiscs show conformations consistent with greater activation than structures in smaller nanodiscs. Additionally, the conformations of ELIC in the larger 11 nm nanodisc are more similar to the conformations sampled in the MD simulations in a lipid bilayer, although they are not the same. Also, the 11 nm nanodisc shows less perturbation of membrane thickness in the manuscript. These findings lead us to suggest that larger nanodiscs may better approximate a lipid bilayer. However, there are likely to be factors other than nanodisc size that could affect protein structure such as direct interactions with the scaffold (in which case the chemistry or structure of the scaffold is important, e.g. SMA) or differential partitioning of lipids into nanodiscs. We have now mentioned these factors in the last paragraph of the discussion. Also, the MD simulations, which have limited sampling, do not fully sample the conformational space of ELIC in a lipid bilayer making us uncertain of the true agonist-bound structure in a lipid bilayer. Therefore, we conclude that further work is needed to understand the impact of nanodiscs on other pLGICs, and, ideally, to determine structures of pLGICs in more native membrane environments.

Reviewers' Comments:

Reviewer #1:

Remarks to the Author:

I am satisfied with the authors revisions and appreciate the time and effort they have taken to carry out this research.

Reviewer #2:

Remarks to the Author:

I appreciate the substantial effort that went into this revision which is much improved and more open with the interpretation of the data. In principle I'd be happy to recommend acceptance.

However, it cannot be accepted at the moment, because neither the dryad repo or the GitHub repo exist or can be accessed - I tried several times.

REVIEWERS' COMMENTS

Reviewer #1 (Remarks to the Author):

I am satisfied with the authors revisions and appreciate the time and effort they have taken to carry out this research.

Reviewer #2 (Remarks to the Author):

I appreciate the substantial effort that went into this revision which is much improved and more open with the interpretation of the data. In principle I'd be happy to recommend acceptance.

However, it cannot be accepted at the moment, because neither the dryad repo or the GitHub repo exist or can be accessed - I tried several times.

We have provided DOIs for both the GitHub repository (which is now archived in Zenodo) and the Dryad repository. The data in Dryad is currently undergoing curation, and will be available within 1-2 weeks.